# Identification of Possible Risk Variants of Familial Strabismus Using Exome Sequencing Analysis

**DOI:** 10.3390/genes12010075

**Published:** 2021-01-10

**Authors:** Joon-Yong An, Jae Ho Jung, Leejee Choi, Eric D. Wieben, Brian G. Mohney

**Affiliations:** 1Department of Biosystems and Biomedical Sciences, College of Health Sciences, Korea University, Seoul 02841, Korea; joonan30@korea.ac.kr; 2Department of Ophthalmology, Seoul National University College of Medicine, Seoul 03080, Korea; jaeho.jung@snu.ac.kr; 3Department of Integrated Biomedical and Life Sciences, College of Health Sciences, Korea University, Seoul 02841, Korea; lizzy723@korea.ac.kr; 4Department of Biochemistry and Molecular Biology, Mayo Clinic, Rochester, MN 55905, USA; wieben.eric@mayo.edu; 5Department of Ophthalmology, Mayo Clinic, Rochester, MN 55905, USA

**Keywords:** strabismus genetics, exome sequencing, strabismus genes, familial strabismus, strabismus genomic analyses

## Abstract

Purpose: To investigate candidate genes associated with familial strabismus and propose a theory of their interaction in familial strabismus associated with early neurodevelopment. Methods: Eighteen families, including 53 patients diagnosed with strabismus and 34 unaffected family members, were analyzed. All patients with strabismus and available unaffected family members were evaluated using whole exome sequencing. The primary outcome was to identify rare occurring variants among affected individuals and investigate the evidence of their genetic heterogeneity. These results were compared with exome sequencing analysis to build a comprehensive genetic profile of the study families. Results: We observed 60 variants from 58 genes in 53 patients diagnosed with strabismus. We prioritized the most credible risk variants, which showed clear segregation in family members affected by strabismus. As a result, we found risk variants in four genes (*FAT3, KCNH2*, *CELSR1*, and *TTYH1*) in five families, suggesting their role in development of familial strabismus. In other families, there were several rare genetic variants in affected cases, but we did not find clear segregation pattern across family members. Conclusion: Genomic sequencing holds great promise in elucidating the genetic causes of strabismus; further research with larger cohorts or other related approaches are warranted.

## 1. Introduction

Strabismus is a common disorder of ocular alignment that affects 3–5% of children in the United States [1,2,3,4]. Although most forms of strabismus in adult age are paralytic, restrictive or otherwise age-dependent [4] the etiology of childhood strabismus is poorly understood. Risk factors for ocular misalignment among a minority of children include premature birth, maternal tobacco exposure and developmental or neurological disorders. More prominently, however, is the central but currently unknown role of genetics in the development of the most prevalent forms of childhood strabismus. A family history of strabismus, for example, is present in 30% of affected individuals [5,6], and twin studies have shown a concordance rate of 70–80% for monozygotic twins and 30–40% for dizygotic twins [5,7,8]. Familial aggregation studies of ocular misalignment have suggested a risk ratio for siblings between three and five [9,10]. Family-based studies have reported the high relative risk for first-degree relatives of an affected proband, implicating the relevance of genetic components underlying strabismus [5,10,11]. Parikh et al. summarized 11 published twin studies of 206 monozygotic and 130 dizygotic twins and found a higher concordance for strabismus in monozygotic twins compared to dizygotic twin pairs [11].

Recent large-scale investigations have demonstrated the genetic roots of strabismus. Genome-wide screening of common strabismus form reported three Mendelian loci (7p22.1, 4q28.3 and 7q31.2) [11,12,13]. The UK Biobank Eye and Vision Consortium, a genome-wide association study of 1345 cases and 65,349 controls, identified a *NPLOC4*–*TSPAN10*–*PDE6G* gene cluster as being associated with an increased risk of strabismus [14]. Common variants within intron 1 of the WRB (tryptophan rich basic protein) were shown to be significantly associated with esotropia and showed paternal transmission bias in paternal inheritance [15].

Recent advances in next-generation sequencing technologies have allowed the identification of disease-causing genes. WES (whole exome sequencing) is a well-established technology for identifying variants within the coding regions, or exons, of known genes, and provides an opportunity to identify causal genes in Mendelian and complex genetic disorders with high sensitivity and specificity. Familial studies using whole exome sequencing (WES) revealed rare variants associated with strabismus or genetic syndromes with strabismus. Gong et al. found a nonsynonymous mutation in paired box 3 (*PAX3*) gene in the two affected individuals with strabismus [16]. Picher-Martel et al. showed a D317V homozygous mutation in TELOE-2 interacting protein 2 (*TTI2*) gene in an individual with global development delay and convergent strabismus [17]. In this study, WES was utilized to detect single nucleotide variants (SNVs) and insertions/deletions (indels) within a cohort of families with two or more members affected with comitant strabismus. We identified rare variants among affected individuals and examined multiple lines of evidence concerning the genetic heterogeneity of identified risk variants and their association with previously described developmental conditions. We then compared these results with exome sequencing analysis to build a comprehensive genetic profile of these families.

## 2. Methods

### 2.1. Sample Collection and Diagnostic Procedures

We prospectively recruited children with familial strabismus, defined as having two or more family member with horizontal, comitant strabismus. The proband and available family members were examined for their visual acuity, angle of deviation at distance (3 m) and at near (1/3 m) by the prism and alternate cover test (PACT), stereopsis, and cycloplegic refraction. Affected members with strabismus were defined as having a horizontal, comitant misalignment of 10 or more prism diopter (PD) at either near or distance in the primary position. Nonaffected individuals were defined as having orthophoria and normal stereopsis. We excluded families with craniosynostosis, molecularly-defined genetic syndromes or developmental delay, and family members who had a neurologic, sensory, or ocular structural disorder as a cause of their strabismus. Families were also excluded if a trio analysis, defined as the proband and both biologic parents, was not available. Among the 98 participants initially recruited, 87 participants remained for downstream analysis, including 53 the affected and 34 the unaffected.

### 2.2. Whole Exome Sequencing

Blood was obtained via venipuncture from all 87 participants and genomic DNA was extracted from whole blood. We used 1 μg of genomic DNA to construct whole genome libraries. Exome libraries were made using the Agilent SureSelect Human All Exon V5+UTR capture kit (Santa Clara, CA, USA) and sequenced with paired-end 100 bp reads on the Illumina HiSeq2000 (San Diego, CA, USA). This approach captures 75 Mb of the genomic sequence primarily comprised of exonic sequence. The GenomeGPS pipeline (Rochester, MN, USA) at Mayo Clinic Rochester was used for alignment and analysis of NextGen sequence data. Sequence reads were generated and aligned to the reference genome hg19 using Novoalign (http://www.novocraft.com) (version 3.09.02). All germline variant calling was jointly called through GATK Haplotype Caller and GenotypeGVCF walkers (https://gatk.broadinstitute.org/) (version 3.4-46-gbc02625). To achieve a high-quality call set, we used multisample calling and Variant Quality Score Recalibrator with training data sets (HapMap3, 1k genome and dbSNP). We discarded genotypes that were likely to be false positives or of poor quality from the list of variants with the following criteria: (1) heterozygous genotypes in the X-chromosome in male samples, (2) genotypes in the Y chromosome in female samples, (3) genotypes covered by fewer than 10 sequence reads, (4) genotypes with a Phred-scaled base quality score lower than 90 and (5) genotypes with indels that are frequently observed across multiple samples. Ancestry of samples was predicted by individual genotype information using the Peddy program [18].

### 2.3. Variant Annotation

We assessed SNVs and indels using the Ensembl VEP software (version 96.7a35428). Our gene annotation was based on the GENCODE v19 which selects the most severe consequence per genetic variant according to the VEP algorithm. For a minor allele frequency of genetic variants, we obtained allele frequency information from the gnomAD database (version r2.1) (https://gnomad.broadinstitute.org/). We selected variants that are predicted as a missense and loss-of-function variants. Loss-of-function variants included frameshift, nonsense and canonical splice site changes, as previously defined in large-scale exome studies [19,20,21]. We evaluated the functional significance of variants using the scale-invariant feature transform (SIFT, version 5.2.2; https://sift.bii.a-star.edu.sg/) and PolyPhen2 (version 2.2.2; http://genetics.bwh.harvard.edu/pph2/) for missense mutations.

## 3. Results

### 3.1. Identification of Risk Genes from WES Data of Strabismus Families

Exome sequencing was performed on 87 individuals of 18 strabismus families with a 53.6X average sequence read depth and 88.5% average call rate (Appendix A). As a result, there were a total of 320,470 unique SNVs (~120,299 SNVs per individual) and 51,072 unique indels (~18,607 indels per individual) found in our WES dataset as shown in Figure 1. Of these, 84,376 SNVs and 3050 indels were located in protein-coding sequences, while 236,094 SNVs and 48,022 indels were found in noncoding sequences.

To identify candidate genes for causing strabismus, we defined causal variants by the following criteria: (1) occurring rarely in the general population, (2) resulting in either a loss of function (LoF; including nonsense, frameshift, and canonical splicing site change), or missense (3) missense variants predicted to be deleterious by either the SIFT or PolyPhen2 algorithm, (4) observed in genes with the probability of being loss-of-function intolerant (pLI) score ≥ 0.9, and (5) characterized in the developmental process (GO:0032502). From 18 families with horizontal and comitant strabismus, we identified 60 candidate variants, including three LoF and 57 missense variants in 58 genes (two genes with multiple variants).

### 3.2. Credible Risk Variants Associated with Familial Strabismus

To assess credible risk variants associated with familial strabismus, we first prioritized a risk variant segregated in multiple family members or occurring in multiple families, possibly suggesting their role in familial strabismus. As a result, we found risk mutations in *FAT* (Family 5 and 9), *KCNH2* (Family 7), *CELSR1* (Family 12), and *TTYH1* (Family 14) (Appendix A). We identified missense mutations in *KCNH2* (Family 7), and *TTYH1* (Family 14), which are essential for cortical layer development in embryonic brains. A neuronal isoform of *KCNH2* plays a critical role in cortical physiology, cognitive function, and neuronal repolarization [22]. *TTYH1* is a chloride anion channel that controls the Notch signaling pathway in neural stem cells [23]. Both father and son in Family 12, affected with intermittent exotropia, had missense mutations in *CELSR1*. *CELSR1* encodes a protein of the Flamingo subfamily, part of the cadherin superfamily, which is widely expressed in the nervous system and plays critical roles in early neurodevelopment, including axon guidance, neuronal migration, and cilium polarity [24]. From a recent study, *celsr3* zebrafish mutants were shown to have a reduced ability for visual tracking and movement as well as a physiological perturbation in GABAergic signaling, implicating *CELSR1* role in the normal development of optic nerve visual pathway circuitry in the inner retina [25].

Multiple cases (*n* = 4) in two families (Family 5 and 9) had a missense variant in *FAT3* (FAT Atypical Cadherin 3) (Figure 2). This gene is essential in neuronal morphogenesis and retina development and controls the development of amacrine cells which project main dendrites into the inner plexiform layer of the retina [26,27].

### 3.3. Genetic Variants with Partial or Weak Evidence for Familial Strabismus

Although credible genetic variants were found in four genes (*FAT3*, *KCNH2*, *CELSR1* and *TTYH1*) in five families, there were several families with various combinations of genetic variants and a lack of clear segregation of affected genes. We thought that these genes hold weak association, but we further explored genes that are known to be associated with the strabismus phenotype (HP:0000486) in human phenotype ontology. As a result, we found three genes (*ARID1B*, *ARNT2*, *COL4A1*) intersected with the HPO strabismus gene list. Family18 in this study had an inherited missense variant in AT-Rich Interaction Domain (*ARID1B*), a member of the SWI/SNF-A chromatin-remodeling complex, previously observed in Coffin-Siris (ORPHA 1465; OMIM 135900) and Nicolaides–Baraitser syndromes (ORPHA:3051; OMIM 601358) [28,29]. In syndromic cases, de novo variants cause ocular abnormalities, including strabismus, as well as a range of other developmental conditions (e.g., hypotonia, seizures). Given the strong selection underlying SNVs in these genes [21,30], inherited variants in the same genes may contribute to the strabismus phenotype without syndromic or strong developmental phenotypes. In Family 6, the three-generation pedigree shows five individuals affected with esotropia. In Family 18, esotropia is present in five members in two generations. *ARID1B* is highly expressed during prenatal brain development, playing a pivotal role in neuronal differentiation and forebrain development. Two members (II.1 and III.1) of five affected individuals in Family 18 harbored an inherited missense variant in *ARNT2*, previously known for the Lachiewicz Sibley syndrome (ORPHA 3157). The gene encodes Aryl hydrocarbon receptor nuclear translocator 2, highly expressed in the lateral geniculate nucleus and superior colliculus during early visual development, and known to regulate the expression of various genes in the mature occipital cortex [31,32]. A previous report showed that a homozygous frameshift in *ARNT2* causes visual anomalies and various developmental delays in children from a highly consanguineous family [33]. In Family 11, an inherited missense variant in *COL4A1* was found in two children (Family 11 II.1, II.2) and one parent (Family 11 I.2), both of whom have intermittent exotropia. A previous report demonstrated that mutations in this gene cause Walker-Warburg Syndrome (ORPHA:899), characterized by ocular dysgenesis, neuronal migration defects, and congenital muscular dystrophy [34]. Viewed together, inherited variants in these syndromic genes may cause a milder form of related phenotypes, which could have been segregated in multiple affected family members without apparent neurological sequelae (Table 1). Such an effect implies that the transmission type (de novo or inherited) may determine the degree and range of phenotypes, as previously proposed in a model of *SCN2A* pathophysiology [35]. In addition, it is noted that affected family members in Family 11 and Family 18 showed variable combinations of genetic variants and genes in affected individuals. A lack of clear segregation could implicate the oligogenic characteristics of strabismus and emphasize the sufficient sample size for further genetic study.

Strabismus is characterized by genetic heterogeneity, presumably with numerous genes involved in its pathophysiology. Given this complexity, causal genes can be expected to play a part in early neurodevelopment and connectivity of brainstem oculomotor neurons [36,37]. Therefore, we examined genes previously associated with optic nerve and retina development. As a result, we observed three genes—*FBN2*, *LRRTM1*, *CTNNA1*—previously described in strabismus families or functionally characterized in model animals (Figure 2).

In Family 11 we found a paternally inherited missense variants in *FBN2* (Fibrillin 2), a structural protein for connective tissue and extracellular calcium-binding microfibrils [38]. A recent study reported severe defects in ocular development in a *Fbn2a* null mice, suggesting its role in early morphogenesis and elastic fiber assembly in developing eyes [39]. The transcriptomics study found that *FBN2A* is highly up-regulated 2 weeks after extraocular muscle surgery and suggests an important role in connective tissues and extraocular muscle adaptation in human eyes.

Two cases in Family 9 (Family 9 I.2, II.1) were affected by a missense variant in *LRRTM1* (Leucine Rich Repeat Transmembrane Neuronal 1). This is a key synaptic adhesion molecule in the emergence of retino-thalamic convergence during early development [40]. An animal study has shown that a genetic deletion of *LRRTM1* resulted in a loss of complex retinal ganglion cell synapses and thus reduced retinal convergence in the visual thalamus [40].

Finally, a missense variant in *CTNNA1* (Catenin α 1) was observed across multiple cases in Family 15, characterized by comitant esotropia (I.2, II.1, and III.1). *CTNNA1* is a part of α-catenins, primarily expressed in neural progenitors in the developing cerebral cortex [41,42] which play a critical role in modulating the optic fissure closure and the organization of neural retina [43]. Previous genetic studies showed that genetic defects in *CTNNA1* may adversely affect the retina and brain during development [44]. Heterozygous missense mutations (p.E307K, p.L318S; p.I431M) in *CTNNA1* were observed in three families with butterfly-shaped pigment dystrophy and characterized by a chemically induced mouse mutant, revealing defects in intercellular adhesion and cytokinesis in the retina [44]. Another WES study reported a heterozygous missense variant (p.G353C) in *CTNNA1* in a patient affected by Leber’s congenital amaurosis with rapid and severe vision loss due to retinal dystrophy [45]. A more recent report found a heterozygous missense variant (p.S322L) in *CTNNA1* in a female patient with retinal dystrophy [46].

## 4. Discussion

Rapid progress in next generation sequencing has accelerated genetic discoveries of human disorders. WES analyses have facilitated the identification of rare genetic variants and risk genes associated with developmental disorders and have provided a comprehensive understanding of the pathophysiology of these disorders. These successful discoveries have enhanced our knowledge of strabismus-associated genes and fostered a protocol for identifying phenotype-genotype relationships [47].

In this study, we systematically evaluated putative risk genes associated with strabismus from 18 families with familial strabismus. Although the size of the sample cohort was insufficient to perform a robust genetic association test for gene-discovery, our analysis prioritized risk variants and genes, implicating the pathophysiology of strabismus in the context of early neurodevelopment. 

Several families in our study had different clinical phenotypes, even though they had the same genetic variation. These results showed the possibility that not only the role of genetic variation in the expression of strabismus, but also the developmental process, the time of ocular misalignment occurrence, and/or environmental factors can affect the type of strabismus. However, the classification of strabismus that used demonstrated the limitation that it was based only on the phenotype rather than based on pathophysiological mechanisms of strabismus. Therefore, we would like to suggest that a new classification and approach to strabismus based on additional biological studies might be addressed in the near future.

## 5. Conclusions

Despite the small sample study, we were able to identify four genes (*FAT3*, *KCNH2*, *CELSR1*, *TTYH1*) with causative associations with strabismus. Further work will require a larger cohort for gene discovery and a complete understanding of its genetic architecture. We also observed genetic heterogeneity in strabismus cases with risk genes occurring in several neurodevelopmental disorders. These results add to the increasing evidence for genetic heterogeneity within familial comitant strabismus, necessitating large sample sizes and less heterogeneous phenotypes for identifying causative variants. We assume that, as with other developmental disorders, different combinations of allelic variants of causative loci putatively confer different risks between individual strabismus cases. Further study, therefore, will need to evaluate the specific contribution of different genetic mutations from copy-number variation to single nucleotide variants. Next-generation genome sequencing approaches, including WES technologies, have significant promise in identifying rare variants associated with the disorder and may fully elucidate the genetic causes of strabismus. 

## Figures and Tables

**Figure 1 genes-12-00075-f001:**
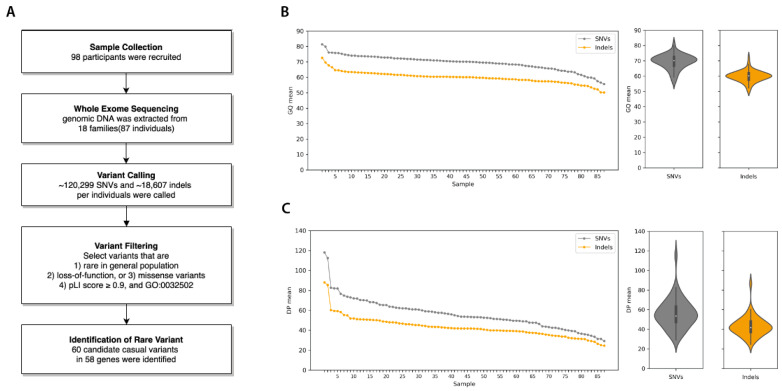
Workflow and quality statistics of familial strabismus exome data. (**A**) Workflow diagram of exome sequence analysis from sample collection to the identification of rare variants in familial strabismus. (**B**) Summary of Genotype Quality (GQ) mean. (**C**) Read Depth (DP) mean per sample. Both quality statistics were measured separately for SNVs (single nucleotide variants) and Indels, for each sample.

**Figure 2 genes-12-00075-f002:**
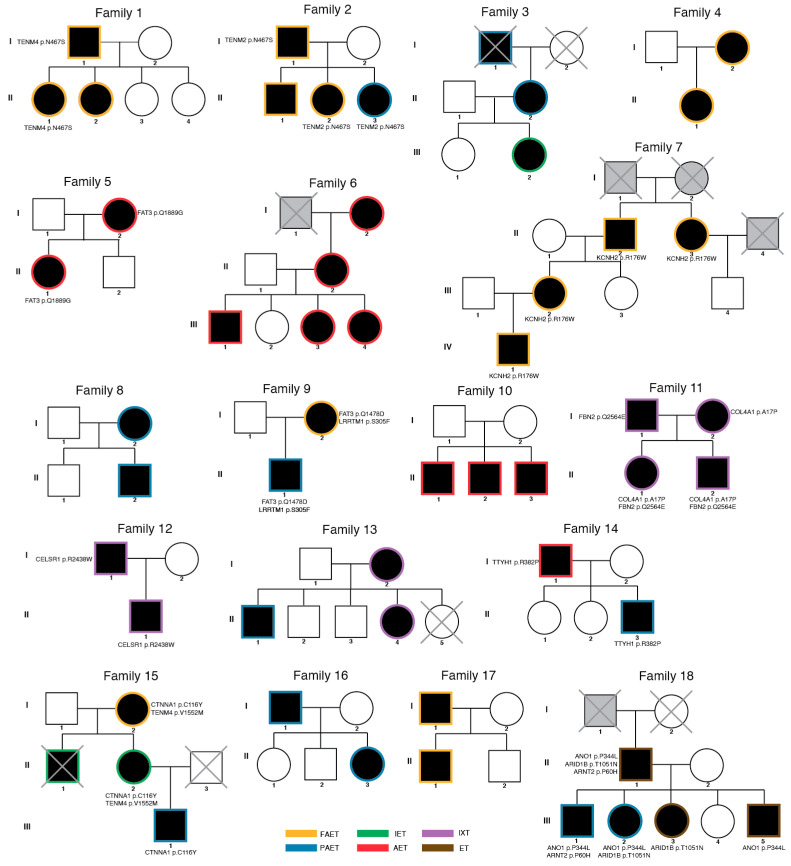
Pedigree information with risk variants. Eighteen families with strabismus are described. Of 98 participants, 87 individuals were analyzed. Participants excluded from the analysis were marked with X. Affected family members are labelled with the black circle (female) and square (male). Unaffected individuals were labelled without the filled color. Individuals with unknown diagnostic status were labelled with grey color. Risk variants are described with their affected gene and amino acid change. All variants are heterozygous. (FAET; Fully accommodative esotropia, PAET; Partially accommodative esotropia, IXT; Intermittent exotropia, ET; Esotropia (unspecified), AET; Acquired esotropia, IET; Infantile esotropia).

**Table 1 genes-12-00075-t001:** List of rare variants in risk genes.

Gene	pLI Score	Variant	Consequence	Ancestry-Matched Allele Frequency	SIFT /PolyPhen2
*ARID1B*	0.999	c.3152C>A (p.T1051N)	missense_variant	8.804 × 10^−6^ (EUR)	D/B
*ARNT2*	0.959	c.179C>A (p.P60H)	missense_variant	Not reported	D/D
*COL4A1*	0.999	c.3712C>T (p.R1238C)	missense_variant	4.20 × 10^−5^ (EUR)	D/D
*FBN2*	0.999	c.7690C>G (p.Q2564E)	missense_variant	7.926 × 10^−5^ (EUR)	D/D
*LRRTM1*	0.951	c.914C>T (p.S305F)	missense_variant	1.761 × 10^−5^ (EUR)	D/D
*CTNNA1*	0.970	c.347G>A (p.C116Y)	missense_variant	2.637 × 10^−5^ (EUR)	D/D
*KCNH2*	0.996	c.526C>T (p.R176W)	missense_variant	6.557 × 10^−4^ (EUR)	D/D
*TTYH1*	0.998	c.1145G>C (p.R382P)	missense_variant	Not reported	B/D
*ANO1*	0.993	c.1031C>T (p.P344L)	missense_variant	Not reported	D/D
*CELSR1*	0.999	c.7312C>T (p.R2438W)	missense_variant	6.408 × 10^−5^ (EUR)	D/D

“pLI score” indicates gene haploinsufficiency defined by the ExAC database. “Ancestry-matched Allele Frequency” was defined as the allele frequency from the gnomAD database for the population matching to each sample’s ancestry (EUR: European, Non-Finnish). Please note that “Not reported” indicate a private mutation that allele frequency is not reported in the database. In the “SIFT/PolyPhen2” column, D (Deleterious) is ‘deleterious_low_confidence/deleterious’ in SIFT and ‘possibly damaging/probably damaging’ in PolyPhen, and B (Benign) is ‘tolerated’ in SIFT and ‘benign’ in PolyPhen.

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
