# Peer review of "Identification of Possible Risk Variants of Familial Strabismus Using Exome Sequencing Analysis"

_genes, 2021, doi:10.3390/genes12010075_

Round 1

Reviewer 1 Report

Introduction is still not adequate. A lot is said about WES but not about the genetics of strabismus.

There are several inconsistencies in the revised version, not explained well by the authors, such as the sample size (figure 1 overall).
Some sentences have been removed and document does not read well after the editing. 

Author Response

Introduction is still not adequate. A lot is said about WES but not about the genetics of strabismus.

  • In the introduction, we have revised the text and added some references for the genetics of strabismus.

There are several inconsistencies in the revised version, not explained well by the authors, such as the sample size (figure 1 overall). Some sentences have been removed and document does not read well after the editing.

  • Thanks for the comment. We found that few texts were not updated so it revised as others.

Reviewer 2 Report

Here are my formal comments.

General Comments:

The introduction describes the many environmental factors that are known to cause childhood strabismus, but that knowledge of genetic causes is lacking. The authors then describe their intent to use WES to look for SNV and indels associated with strabismus.

Methods: While the number of participants was small (87 in total: 53 affected and 34 not affected), the authors screened their participants well and excluded anyone without both biological parents being available for the study, i.e. trio analysis (proband and parents). This is important in order to make the connection between the trait and if it was potentially inherited as opposed to a unique variation to the proband only. The data generation, calling, alignment and cleaning are following normal methods and established programs. The criteria for discarding heterozygous genotypes, X or Y chromosome, low read depth and quality are appropriate and reasonable. The selection of only missense and loss of function variants as described in the paper is appropriate and well referenced against other studies.

Results: The pedigrees in figure 2 are short in that they only cover two or three generations (due to practical reasons of data collection), but these are long enough to support reasonable speculation about the inheritance of the variation identified via WES. The supporting evidence for the variants identified in this study for having or potentially having a relationship with various forms of strabismus is compelling. While not statistically definitive, due to a small study sample that the authors address in the text, the results are useful for those working on this particular trait and to those that may be looking for a method of studying other heterozygous traits.

Discussion: The discussion, and results are much improved due to the comments of Reviewer 3, which lead to a more focused paper, as evidenced by the data added to and removed from the first version of this work, as well as the change in the title. It shows variations that may be potentially associated with phenotypic traits can be found helping to narrow down the large number of variants that could be associated to a manageable number to begin with. Some of these may not work out, but the authors did supply supporting evidence for the literature to strengthen the case for a closer examination of several of the variants identified in this study.

Specific Comments:

Abstract: Line 28: Chang to “FAT3” Methods: Line 107: make it clear that the 320,470 SNVs and 51,072 were the number of unique ones found. According to Supp Table 1 a lot more were found. I take it the number reported here is the unique number. If so then make it clear in the sentence.

Author Response

General Comments:

The introduction describes the many environmental factors that are known to cause childhood strabismus, but that knowledge of genetic causes is lacking. The authors then describe their intent to use WES to look for SNV and indels associated with strabismus.

Methods: While the number of participants was small (87 in total: 53 affected and 34 not affected), the authors screened their participants well and excluded anyone without both biological parents being available for the study, i.e. trio analysis (proband and parents). This is important in order to make the connection between the trait and if it was potentially inherited as opposed to a unique variation to the proband only. The data generation, calling, alignment and cleaning are following normal methods and established programs. The criteria for discarding heterozygous genotypes, X or Y chromosome, low read depth and quality are appropriate and reasonable. The selection of only missense and loss of function variants as described in the paper is appropriate and well referenced against other studies.

Results: The pedigrees in figure 2 are short in that they only cover two or three generations (due to practical reasons of data collection), but these are long enough to support reasonable speculation about the inheritance of the variation identified via WES. The supporting evidence for the variants identified in this study for having or potentially having a relationship with various forms of strabismus is compelling. While not statistically definitive, due to a small study sample that the authors address in the text, the results are useful for those working on this particular trait and to those that may be looking for a method of studying other heterozygous traits.

Discussion: The discussion, and results are much improved due to the comments of Reviewer 3, which lead to a more focused paper, as evidenced by the data added to and removed from the first version of this work, as well as the change in the title. It shows variations that may be potentially associated with phenotypic traits can be found helping to narrow down the large number of variants that could be associated to a manageable number to begin with. Some of these may not work out, but the authors did supply supporting evidence for the literature to strengthen the case for a closer examination of several of the variants identified in this study.

  • Thanks for the comment. In the introduction, we cited the genetic studies of strabismus using genetic screening, literature review, or genome-wide association studies (GWAS) methods. Our study aims to utilize whole exome sequencing (WES) to find rare variants associated with strabismus. So, we added some reference for the recent WES studies for strabismus.
  • Thanks for your comment. As you pointed out, there is limitation in data collection and small sample size. We hope to continue to increase our sample size in next few years and apply genome-wide association method for statistical evaluation.

Specific Comments:

Abstract: Line 28: Change to “FAT3”

  • We revised this.

Methods: Line 107: make it clear that the 320,470 SNVs and 51,072 were the number of unique ones found. According to Supp Table 1 a lot more were found. I take it the number reported here is the unique number. If so then make it clear in the sentence.

  • The number comes from the total SNVs and indels from our WES data. So, we updated the numbers for each individual as we refer in the text. To make it clear, we have revised our text and Figure 1.

Reviewer 3 Report

this is an interesting, well-written study.

few suggestions:

strabismus' description should be improved for example:

- presence of eso- or exotropia,

- presence of oblique muscles hypo-hyperfunction and alphabetic pattern,

- presence of high AC/A ratio

it would be interesting to know whether one or more of previous findings might be correlated with one or more identified genes 

Author Response

This is an interesting, well-written study.

Few suggestions:

strabismus' description should be improved for example:

- presence of eso- or exotropia,

- presence of oblique muscles hypo-hyperfunction and alphabetic pattern,

- presence of high AC/A ratio

It would be interesting to know whether one or more of previous findings might be correlated with one or more identified genes

  • Thanks for your suggestion. We have made changes in Figure 2 by adding colors for (FAET), Fully accommodative esotropia, PAET (Partially accommodative esotropia), ​IXT (Intermittent exotropia), ​ET (Esotropia (unspeicified)), ​AET (Acquired esotropia), and IET (Infantile esotoropia). We also added the sentences in Line 281 as following:
    “Several families in our study had different clinical phenotypes, even though they had the same genetic variation. These results showed the possibility that not only the role of genetic variation in the expression of strabismus, but also the developmental process, the time of occurrence, and/or environmental factors can affect the type of strabismus. However, the classification of strabismus that we are using demonstrated the limitation that it was based only on the phenotype rather than based on pathophysiological mechanisms of strabismus. Therefore, we would like to suggest that a new classification and approach to strabismus based on additional biological studies might be addressed in near future.”

Reviewer 4 Report

Authors An et al. used whole exome sequencing (WES) to investigate potential genetic causes of strabismus. In the study, 87 individuals were analysed, 53 affected. In this study, 60 variants in 58 genes associated with neurodevelopment, rare syndromic disease or optic nerve development were identified.

However, there is no proof that these genes can be evaluated as risk factors and are not only a rare/private variant. Therefore I suggest to change title from “Identification of risk genes of familial strabismus using an exome sequencing analysis“ to “Identification of rare variants in families with strabismus using an exome sequencing analysis“

Major comments

Please add numbers of patients and unaffected family members into Methods section, as these numbers are in abstract only. Additionally, I suggest to visualise which family member was analysed in Figure 2, pedigrees.

Please use more than two prediction programs (SIFT and PolyPhen2). For example, MutPred2, SNP&GO, Mutation Taster etc.). You should also use ACMG standards to classify observed variants (online tool available at https://www.medschool.umaryland.edu/Genetic_Variant_Interpretation_Tool1.html/)

Table 1 – column 1 “Gene” should be in italics, “variant” should be genome position or description on DNA level, not at protein levels, also provide reference sequence for each gene. Number of protein domains can be omitted.  

Minor comments

Line 45: …identified a NPLOC4–TSPAN10–PDE6G gene cluster … - please use italics

Line 46: Common variants within intron 1 of the WRB… - please use italics

Line 176-177, line 203: please check the font

Author Response

Authors An et al. used whole exome sequencing (WES) to investigate potential genetic causes of strabismus. In the study, 87 individuals were analysed, 53 affected. In this study, 60 variants in 58 genes associated with neurodevelopment, rare syndromic disease or optic nerve development were identified.

However, there is no proof that these genes can be evaluated as risk factors and are not only a rare/private variant. Therefore I suggest to change title from “Identification of risk genes of familial strabismus using an exome sequencing analysis“ to “Identification of rare variants in families with strabismus using an exome sequencing analysis“

  • Thanks for the comment. We have updated our text as suggest.

Major comments

Please add numbers of patients and unaffected family members into Methods section, as these numbers are in abstract only. Additionally, I suggest to visualise which family member was analysed in Figure 2, pedigrees.

  • Thanks for your suggestion. We have revised Figure 2 and its legend to include the number of family members used in the analysis. Also, we marked members not used in the analysis.

Please use more than two prediction programs (SIFT and PolyPhen2). For example, MutPred2, SNP&GO, Mutation Taster etc.). You should also use ACMG standards to classify observed variants (online tool available at https://www.medschool.umaryland.edu/Genetic_Variant_Interpretation_Tool1.html/)

  • Thanks for your suggestion. We have included additional variant prediction and ACMG classification results for variants. For variant prediction, we added three prediction tools, including SNPs&GO, Mutation Taster, and CADD. As predicted by SIFT and PolyPhen2, all variants are predicted as “deleterious (or damaging)” by at least one tool though few variants have unmatched prediction due to a prediction algorithm. Including various prediction results would provide more comprehensiveness for variant interpretation. Using the ACMG guideline, we were able to classify observed variants and summarize the evidence categories in Table S2.

Table 1 – column 1 “Gene” should be in italics, “variant” should be genome position or description on DNA level, not at protein levels, also provide reference sequence for each gene. Number of protein domains can be omitted.  

  • Thanks for your suggestion. We have revised Table 1.

Minor comments

Line 45: …identified a NPLOC4–TSPAN10–PDE6G gene cluster … - please use italics

Line 46: Common variants within intron 1 of the WRB… - please use italics

Line 176-177, line 203: please check the font

  • We revised these.

Round 2

Reviewer 1 Report

I still think the authors can improve the introduction further and include recent articles, such as:

Linkage analysis identifies an isolated strabismus locus at 14q12 overlapping with FOXG1 syndrome region.

Ye XC, Roslin NM, Paterson AD, Lyons CJ, Pegado V, Richmond P, Shyr C, Fornes O, Han X, Higginson M, Ross CJ, Giaschi D, Gregory-Evans C, Patel MS, Wasserman WW.J Med Genet. 2020 Nov 30:jmedgenet-2020-107226. doi: 10.1136/jmedgenet-2020-107226. Online ahead of print.PMID: 33257509   The Role of Heredity and the Prevalence of Strabismus in Families with Accommodative, Partial Accommodative, and Infantile Esotropia. Çorak EroÄŸlu F, Oto S, Åžahin FÄ°, Terzi Y, Özer Kaya Ö, Tekindal MA.Turk J Ophthalmol. 2020 Jun 27;50(3):143-150. doi: 10.4274/tjo.galenos.2019.49204.PMID: 32631000

Author Response

Thank you. I have revised all the comments.

Reviewer 4 Report

Authors had modified the manuscript according to my suggestions. No other comments.

Author Response

Thank you. I have revised all the comments.

This manuscript is a resubmission of an earlier submission. The following is a list of the peer review reports and author responses from that submission.

Round 1

Reviewer 1 Report

Introduction must include information on the previous studies regarding the genes and genetic factors Involved in strabismus. 

In the results section, the pathways mentioned to be involved In strabismus should be Described and  explained better.

Author Response

  • Introduction must include information on the previous studies regarding the genes and genetic factors Involved in strabismus.
    We appreciate the comments of the reviewer and have added the latest large-scale genetic studies of strabismus and revised the text as follows:
    The UK Biobank Eye and Vision Consortium, a genome-wide association study of 1345 cases and 65349 controls, identified the NPLOC4–TSPAN10–PDE6G gene cluster as being associated with an increased risk of strabismus. Common variants within intron 1 of the WRB were shown to be significantly associated with esotropia and showed transmission bias in paternal inheritance.

  • In the results section, the pathways mentioned to be involved In strabismus should be Described and explained better.
    Thank you for your comment. We received a similar question from Reviewer 3 as to the network analysis and gene selection. We revised the text of this section for further clarification.

Reviewer 2 Report

General Comments:

Supplementary Table 1 was missing.

Specific Comments:

Abstract:

Line 33: I think this should be five genes as KCNH2 should be included here.

Materials and Methods:

For all databases and software used state the version number used as was done with Ensemble, i.e. version numbers for gnomeAD, BioGrid, CytoScape etc.

Results:

Line 127: The supplementary table 1 is missing.

Line 206: For consistency the order of affected individuals should be (I.2, II.1, and III.1) please correct.

Line 210: Should the 35 be inside of square brackets, i.e. [35]? Is it a refence or a typo?

Line 270: comparing the gene list here to that of Ye et al I can find three genes that overlap. The two you found ARID1B and COL4A1, but also ARID1A. However, ARID1A is in the permissive strabismus data set, whereas the two you have listed are in the stringent strabismus data set. Please clarify.

Lines 271 to 273: This is unclear if you are trying to indicate that your analysis found an additional 13 genes that were not in the Ye at al list, or that your found 13 more genes from your network analysis, some of which are in the Ye list and some are not, e.g. CLIP1 is in your list and in Ye. Please clarify your meaning here.

Figures:

Figure 1. Is the number of samples shown in B and C 91 or 112? You collected 112 samples but only extracted from 91. Please clarify. It would help to put numbers on the x-axis of the graphs so we could see the number of samples it represents and not just a single tick mark on the x-axis.

Tables:

Table 2: The CTNNA1 gene is associated with family 15 not 18, please correct.

Author Response

  • Supplementary Table 1 was missing.
    Thank you for your comment. We uploaded Supplementary Table 1.

Abstract:

  • Line 33: I think this should be five genes as KCNH2 should be included here.
    We modified the text as suggested:
    “Among 60 variants from 58 genes observed, five (KCNH2, TTYH1, ANO1, RIMS2, and CELSR1) are associated with early neurodevelopment”

Materials and Methods:

  • For all databases and software used state the version number used as was done with Ensemble, i.e. version numbers for gnomeAD, BioGrid, CytoScape etc.
    We appreciate the reviewers request and revised the text to include the version of the databases and software we used. Please find them in Materials and Methods:
    - STRING interaction database (version 11.0)
      - CytoScape (version 3.8.0)
      - BioGrid database (version 3.5.170)
      - Novoalign (version 3.09.02)
      - GATK (version 3.4-46-gbc02625)
      - gnomAD (version r2.1)
      - SIFT (version 5.2.2)
      - PolyPhen2 (version 2.2.2)

Results:

  • Line 127: The supplementary table 1 is missing.
    We revised the text to add Supplementary Table 1.

  • Line 206: For consistency the order of affected individuals should be (I.2, II.1, and III.1) please correct.
    As suggested, we revised the text as following:
    Finally, a missense variant in CTNNA1 (Catenin Alpha 1) was observed across multiple cases in Family 15, characterized by comitant esotropia (I.2, II.1, and III.1).

  • Line 210: Should the 35 be inside of square brackets, i.e. [35]? Is it a reference or a typo?
    This typo was deleted from the text.

  • Line 270: comparing the gene list here to that of Ye et al I can find three genes that overlap. The two you found ARID1B and COL4A1, but also ARID1A. However, ARID1A is in the permissive strabismus data set, whereas the two you have listed are in the stringent strabismus data set. Please clarify.
    As suggested by the reviewer, we clarified that ARID1A is in the permissive gene list, whereas ARID1B and COL4A1 are in the stringent gene list.

  • Lines 271 to 273: This is unclear if you are trying to indicate that your analysis found an additional 13 genes that were not in the Ye at al list, or that your found 13 more genes from your network analysis, some of which are in the Ye list and some are not, e.g. CLIP1 is in your list and in Ye. Please clarify your meaning here.
    We appreciate the reviewer’s comment. In this section we tried to indicate that we found more genes from our network analysis. We discovered that 3 genes are not in the strabismus gene list of Ye et al. 2014 and revised the text appropriately.

Figures:

  • Figure 1. Is the number of samples shown in B and C 91 or 112? You collected 112 samples but only extracted from 91. Please clarify. It would help to put numbers on the x-axis of the graphs so we could see the number of samples it represents and not just a single tick mark on the x-axis.
    We agree with the reviewer’s comments. We updated the number of participants we recruited and the number of samples that underwent the exome sequencing analysis. The numbers are now updated in the workflow.

Tables:

  • Table 2: The CTNNA1 gene is associated with family 15 not 18, please correct.
    We corrected the family number as requested.

Reviewer 3 Report

Review of: Genetic Investigation of Familial Strabismus: An 2 Exome Sequencing and Systems Approach

In summary: some data are interesting but the reasoning often makes little sense.

The authors recruited "children with familial strabismus, defined as having two or more family member with horizontal, comitant strabismus". If pedigrees are small, then this strategy biases towards a dominant mode of inheritance. With a recessive mode, only information from a larger pedigree may show further affected persons. At least the proband as well as both biological parents needed to be available. Altogether data from 18 families were available. The pedigrees (with the status of individuals wrt to strabismus, gender, and variants of genes implied causing the trait) are given in Fig 2. Obviously inheritance patterns in all families are consistent with a dominant mode of inheritance.

A whole exome sequencing study was then performed. With this, by definition, data on exons of genes are available, but not on regulating sequences or on copy number variation. The authors report in some families segregation of variants in exons of genes previously identified as likely causal for strabismus (ARID1A, ARID1B, ARNT2, COL4A1).

Interestingly only in few families (5, 7, 9, 11, 12, 14, and 18) variant alleles cosegregate with the trait. With families 11 and 18, two or three loci seem to segregate in a complex fashion. Hence, only families 5, 7, 9, 12, and 14 show a simple pattern of inheritance of alleles of the loci FAT3 (families 5 and 9), KCNH2 (family 7), CELSR1 (family 12), TTYH1 (family 15) that cosegregate with the trait. Hence, these loci (genes) are the strongest candidates for a causal association, especially FAT3, which is found in two families.

Is there any molecular evidence suggesting that the inheritance patterns of these four genes may be dominant? A dominant inheritance pattern is actually rather rare in genetics, since nonsense and many missense mutations usually lead to loss of function. In heterozygotes, the wildtype allele can often compensate completely for the mutant. Dominance occurs eg in signaling molecules, which may be rendered constitutively active through a mutation.

The case for the other genes mentioned by the authors to cause strabismus is very weak, given the inheritance pattern observed in this study; evidence comes mainly from other studies. Why do variants of the genes ARID1A, ARID1B, ARNT2, COL4A1 not co-segregate with the trait, if they really are causally linked to strabismus? The authors may argue (and seem to do so) that strabismus is obviously influenced by many loci and alleles that are actually inherited should be comparatively mild, as de novo mutations of these four genes may cause syndromes that reduce fitness, ie the chance of transmitting alleles to the next generation. Nevertheless, an allele of a major locus contributing to strabismus should co-segregate with the trait. Hence, this study rather seems to weaken the association of these four genes with the trait.

Another part of the study, the network analysis, really relies on the genes identified in the first part to be causal. In my opinion, only four genes in the first part may be causally related to the trait. And with the one exception, the alleles co-segregate only in a single family. Hence, I cannot see much sense in further analyses.

This brings me to another point of criticism: the language or the reasoning is in parts very difficult to understand, eg:

line 45: " Although most adult-onset forms of strabismus are paralytic, restrictive, or otherwise age-dependent,.." What is meant with "restrictive" in this context? If a trait is "adult-onset" it is clearly age-dependent, ie the onset depends on the individual being of adult age. This sentence makes no sense.

line 52: "Familial aggregation studies of ocular misalignment have suggested a risk ratio for siblings of between age 3 and 5." Maybe the authors mean that a child between three and five years old has a higher risk of strabismus, if his/her sibling has it.

line 55: What are "fusion centers" in this context?

What the authors write about "whole exome sequencing" (WES) makes little sense. With WES, of course, only alleles within exons can be detected. If such alleles actually cause a certain disease, the technique may contribute to the detection of an association between an allele and a trait. If the trait is caused by mutations in eg regulatory sequences or copy number variants, WES cannot help. Why write: "[WES] provides an opportunity to identify causal genes in Mendelian and complex genetic disorders with high sensitivity and specificity"? Sensitivity and specificity make sense in the context of binary tests, but WES is not a test. It is a method to relatively quickly sequence a, hopefully relevant, part of the genome. The following is similarly useless: "These successful discoveries have positioned WES as the gold standard for clinical genetic testing ...": WES is not a test!

Other nonsensical sentences are eg the subheading: "Systems biology approaches for genes with familiar strabismus". No gene can have the trait "strabismus", whether or not it is "familiar".

Or the following: "We assume that, like other developmental disorders, different genetic profiles putatively confer different risks between individual strabismus cases." Here a "with" seems to be missing: "like WITH other developmental disorders". But what do the authors mean with "genetic profile", maybe combinations of allelic variants of causative loci?

What is meant with: "Further study, therefore, will need to evaluate the specific contribution of different genetic components from copy-number variation to small variants – SNVs and indels"? Do the authors mean "mutants" with "genetic components"; copy-number variants would fall into this category. What are "small variants" in this context, maybe single basepair variants or polym

Author Response

In summary: some data are interesting but the reasoning often makes little sense. The authors recruited "children with familial strabismus, defined as having two or more family member with horizontal, comitant strabismus". If pedigrees are small, then this strategy biases towards a dominant mode of inheritance. With a recessive mode, only information from a larger pedigree may show further affected persons. At least the proband as well as both biological parents needed to be available. Altogether data from 18 families were available. The pedigrees (with the status of individuals wrt to strabismus, gender, and variants of genes implied causing the trait) are given in Fig 2. Obviously inheritance patterns in all families are consistent with a dominant mode of inheritance.

A whole exome sequencing study was then performed. With this, by definition, data on exons of genes are available, but not on regulating sequences or on copy number variation. The authors report in some families segregation of variants in exons of genes previously identified as likely causal for strabismus (ARID1A, ARID1B, ARNT2, COL4A1).

Interestingly only in few families (5, 7, 9, 11, 12, 14, and 18) variant alleles cosegregate with the trait. With families 11 and 18, two or three loci seem to segregate in a complex fashion. Hence, only families 5, 7, 9, 12, and 14 show a simple pattern of inheritance of alleles of the loci FAT3 (families 5 and 9), KCNH2 (family 7), CELSR1 (family 12), TTYH1 (family 15) that cosegregate with the trait. Hence, these loci (genes) are the strongest candidates for a causal association, especially FAT3, which is found in two families.

  • Is there any molecular evidence suggesting that the inheritance patterns of these four genes may be dominant? A dominant inheritance pattern is actually rather rare in genetics, since nonsense and many missense mutations usually lead to loss of function. In heterozygotes, the wildtype allele can often compensate completely for the mutant. Dominance occurs eg in signaling molecules, which may be rendered constitutively active through a mutation.
    We appreciate the perspective of the reviewer. We do not have further evidence of these four genes at the molecular level. However, from the population exome dataset, these four genes are highly haploin sufficient. Thus, heterozygous mutations may impact their function in developmental processes. Lek et al. (2016 Nature) provides the metrics of gene haploinsufficiency, scaled from 0 to 1 (called pLI scores (probability of Loss of function Intolerance). Genes with a pLI score ≥ 0.9 are called “haploinsufficient genes”. Four genes have a pLI score:
    - FAT3: 0.999
      - KCNH2: 0.996
      - CELSR1: 0.999
      - TTYH1: 0.998
    For these reasons, we are fairly confident that rare heterozygous variants in these genes would cause molecular changes in key signaling pathways related to human development
    .

  • The case for the other genes mentioned by the authors to cause strabismus is very weak, given the inheritance pattern observed in this study; evidence comes mainly from other studies. Why do variants of the genes ARID1A, ARID1B, ARNT2, COL4A1 not co-segregate with the trait, if they really are causally linked to strabismus? The authors may argue (and seem to do so) that strabismus is obviously influenced by many loci and alleles that are actually inherited should be comparatively mild, as de novo mutations of these four genes may cause syndromes that reduce fitness, ie the chance of transmitting alleles to the next generation. Nevertheless, an allele of a major locus contributing to strabismus should co-segregate with the trait. Hence, this study rather seems to weaken the association of these four genes with the trait. Another part of the study, the network analysis, really relies on the genes identified in the first part to be causal. In my opinion, only four genes in the first part may be causally related to the trait. And with the one exception, the alleles co-segregate only in a single family. Hence, I cannot see much sense in further analyses.
    We agree with Reviewer that only four genes in the first part are causally related to strabismus. However, we found it relevant to discuss whether genes with a weak association show some functional patterns in developmental pathways. We fully acknowledge that confirmatory studies are necessary and are cautious on suggesting a weak association. Thus, we added the text for further clarification as following:
    “However, due to the small sample size and resulting discovery of only four genes (FAT3, KCNH2, CELSR1, TTYH1) with a causative association, additional studies are warranted to confirm these findings.

  • This brings me to another point of criticism: the language or the reasoning is in parts very difficult to understand, eg: line 45: " Although most adult-onset forms of strabismus are paralytic, restrictive, or otherwise age-dependent,.." What is meant with "restrictive" in this context? If a trait is "adult-onset" it is clearly age-dependent, ie the onset depends on the individual being of adult age. This sentence makes no sense.
    We appreciate the reviewer’s thoughts. In this sentence we are contrasting acquired strabismus found in adults with that observed in children, the latter of which is more likely associated with premature birth, developmental and syndromic disorders. “Restrictive” in this sentence is most commonly refereeing to thyroid-associated ophthalmopathy (TAO). Most of TAO-related strabismus occur in the adult age group.

  • line 52: "Familial aggregation studies of ocular misalignment have suggested a risk ratio for siblings of between age 3 and 5." Maybe the authors mean that a child between three and five years old has a higher risk of strabismus, if his/her sibling has it.
    We agree with the reviewer and have corrected this sentence. These numbers represented a risk ratio of strabismus based on familial aggregation studies. In addition, we updated this section with the following reference:
    Chen et al., Prevalence, incidence and risk factors of strabismus in a Chinese population-based cohort of preschool children: the Nanjing Eye Study. Br J Ophthalmol. 2020 Aug 22:bjophthalmol-2020-316807. doi:10.1136/bjophthalmol-2020-316807

  • line 55: What are "fusion centers" in this context?
    Fusion centers include a convergent center at the rostral–dorsal midbrain and a divergence center that, based on acute onset of concomitant esotropia related to tumors, is likely situated in the hindbrain.

  • What the authors write about "whole exome sequencing" (WES) makes little sense. With WES, of course, only alleles within exons can be detected. If such alleles actually cause a certain disease, the technique may contribute to the detection of an association between an allele and a trait. If the trait is caused by mutations in eg regulatory sequences or copy number variants, WES cannot help. Why write: "[WES] provides an opportunity to identify causal genes in Mendelian and complex genetic disorders with high sensitivity and specificity"? Sensitivity and specificity make sense in the context of binary tests, but WES is not a test. It is a method to relatively quickly sequence a, hopefully relevant, part of the genome. The following is similarly useless: "These successful discoveries have positioned WES as the gold standard for clinical genetic testing ...": WES is not a test!
    We agree with Reviewer’s comment and revised our text as following: “These successful discoveries have enhanced our knowledge of strabismus-associated genes”

  • Other nonsensical sentences are eg the subheading: "Systems biology approaches for genes with familiar strabismus". No gene can have the trait "strabismus", whether or not it is "familiar".
    Thanks for the comment. We agree with Reviewer’s comment and revised our text as following: “Systems biology approaches for genes found in individuals with strabismus”

  • Or the following: "We assume that, like other developmental disorders, different genetic profiles putatively confer different risks between individual strabismus cases." Here a "with" seems to be missing: "like WITH other developmental disorders". But what do the authors mean with "genetic profile", maybe combinations of allelic variants of causative loci?
    We appreciate the kind suggestion. As indicated, the word “genetic profiles” means combinations of allelic variants of causative loci. Thus, we revised the text as following:
    “like with other developmental disorders, different combinations of allelic variants of causative loci putatively confer different risks between individual strabismus cases”

  • What is meant with: "Further study, therefore, will need to evaluate the specific contribution of different genetic components from copy-number variation to small variants – SNVs and indels"? Do the authors mean "mutants" with "genetic components"; copy-number variants would fall into this category. What are "small variants" in this context, maybe single basepair variants or polym
    Our intent was as the reviewer suggested. For clarification, we modified our text as following:
    “Further study, therefore, will need to evaluate the specific contribution of different genetic mutations from copy-number variation to single nucleotide variants.”

Round 2

Reviewer 3 Report

2nd Review of: Genetic Investigation of Familial Strabismus: An 2 Exome Sequencing and Systems Approach

The authors eliminated many problematic and sometimes implausible phrases. Generally, there is one central and big problem remaining: I do not agree with the authors that they show *causality*, rather they only show *association*. Furthermore, the *inheritance mode* requires more care. Correcting these problems will leave little substance.

The authors write: "To identify candidate genes for causing strabismus, we defined causal variants by the following criteria: 1) occurring rarely in the general population, 2) resulting in either a loss of function (LoF; including nonsense, frameshift, and canonical splicing site change), or missense 3) missense variants predicted to be deleterious by either SIFT or PolyPhen2 algorithm, 4) observed in genes with the probability of being loss-of-function intolerant (pLI) score ≥ 0.9, and 5) characterized in the developmental process (GO::0032502)."

Such a strategy only identifies *candidate* genes (as correctly written) but it does not identify *causal* variants (as incorrectly written). Given that there are thousands of genes in the genome, some variants may fulfill these criteria by chance (which corresponds to the familiar multiple test problem). Only a molecular analysis, either in vitro or in vivo, using animal models, would really identify a causal variant. Furthermore, segregation in the affected families alone is insufficient: rather mutant alleles should co-segregate with the phenotype (in a plausible fashion). For a mutant allele that does not co-segregate take eg family 2: the allele TENM2_p.N467S is found in heterozygous form in the affected father and the two affected daughters, but not in the affected son. This inheritance pattern speaks *against* a causal association, as, even though the variant segregates in an affected family, its inheritance pattern is random with respect to the phenotype. The same can be observed eg in family 1 and family 15. Sometimes inheritance patterns are not so clearly inconsistent, but nevertheless are inconsistent with Mendelian inheritance of a (dominant, codominant, or recessive) monogenic inheritance, eg in family 6. Generally, it seems that, with more than minimal family sizes (ie with families with more than three members), deviations from monogenic Mendelian inheritance can often be observed. Of the families with at least four members, only in families 5, 7, and 11 the inheritance pattern is consistent with a monogenic Mendelian inheritance. In all other families, a case can be made *against* the reported variants being causal, even though they obviously fulfill the criteria of the authors above. The authors should therefore concentrate exclusively on those genes with alleles consistent with Mendelian inheritance: FAT3, KCNH2, and TTYH1. It makes no sense to include the other genes in further analysis (eg gene expression and pathway). In particular and as an example, it makes no sense to follow up on FBN2, since the mother is also affected but does not have the allele. Thus this family and therefore the whole study provides nearly no evidence (positive or negative) for association of FBN2 with strabismus.

Note that criteria 1), 2), 3), and 4) above are rather associated with a recessive inheritance pattern. But the variants co-segregating with the phenotypes in families 5, 7, and 11 have dominant inheritance patterns. The authors should therefore include, their answer to my question in some way in the article, which I copy into here:

We appreciate the perspective of the reviewer. We do not have further evidence of these four genes at the molecular level. However, from the population exome dataset, these four genes are highly haploin sufficient. Thus, heterozygous mutations may impact their function in developmental processes. Lek et al. (2016 Nature) provides the metrics of gene haploinsufficiency, scaled from 0 to 1 (called pLI scores (probability of Loss of function Intolerance). Genes with a pLI score ≥ 0.9 are called “haploinsufficient genes”. Four genes have a pLI score:
- FAT3: 0.999
  - KCNH2: 0.996
  - CELSR1: 0.999
  - TTYH1: 0.998
For these reasons, we are fairly confident that rare heterozygous variants in these genes would cause molecular changes in key signaling pathways related to human development
.

From the authors' information and from NCBI, the action of the three consistent candidate genes are as follows: KCNH2 and TTYH1 are ion channels, which may well show haplo-insufficiency. FAT3 is a cadherin. Since cadherins interact with other proteins, a variant may well show dominant action by abnormally binding to its interaction partner. Hence, these three genes to well be causal; at least they are not inconsistent with causality as with the other genes in this study. This is all the authors are able to say from their data, as far as I can see.

In summary: the authors have indications that the genes FAT3, KCNH2, and TTYH1 are co-inherited with strabismus in an autosomal dominant fashion. Their molecular functions are consistent with dominant inheritance as is the haploinsufficiency score the authors provide. For the other genes, of which some are thought to be involved in strabismus from earlier studies, inheritance patterns are not consistent with a Mendelian inheritance scheme, which weakens their association with strabismus. These should be left out of any further analysis.

Specifics:

l 176: "..., which could have segregated" (or mayby: "which could have been segregating")